EMBO
Molecular Medicine

# The global emergence of a novel *Streptococcus suis* clade associated with human infections

Xingxing Dong[1,2,†], Yanjie Chao[3,†] , Yang Zhou[4,5], Rui Zhou[4], Wei Zhang[6], Vincent A. Fischetti[7], Xiaohong Wang[1], Ye Feng[8,*] & Jinquan Li[1,4,7,**]

## Abstract

***Streptococcus suis***, a ubiquitous bacterial colonizer in pigs, has recently extended host range to humans, leading to a global surge of deadly human infections and three large outbreaks since 1998. To better understand the mechanisms for the emergence of cross-species transmission and virulence in human, we have sequenced 366 *S. suis* human and pig isolates from 2005 to 2016 and performed a large-scale phylogenomic analysis on 1,634 isolates from 14 countries over 36 years. We show the formation of a novel human-associated clade (HAC) diversified from swine *S. suis* isolates. Phylogeographic analysis identified Europe as the origin of HAC, coinciding with the exportation of European swine breeds between 1960s and 1970s. HAC is composed of three sub-lineages and contains several healthy-pig isolates that display high virulence in experimental infections, suggesting healthy-pig carriers as a potential source for human infection. New HAC-specific genes are identified as promising markers for pathogen detection and surveillance. Our discovery of a human-associated *S. suis* clade provides insights into the evolution of this emerging human pathogen and extend our understanding of *S. suis* epidemics worldwide.

**Keywords** human pathogen; population genomics; ST1; ST7; *Streptococcus suis*

**Subject Categories** Evolution & Ecology; Microbiology, Virology & Host Pathogen Interaction

## Introduction

Emerging human pathogens of animal origin have become a major threat to global health, illustrated by the current pandemic of bat-derived SARS-CoV-2. Another important emerging human pathogen is *Streptococcus suis*, which asymptomatically colonizes nearly all pigs (Wertheim *et al*, 2009), but has caused three deadly outbreaks in humans within the last decades (Chen *et al*, 2007; Willemse *et al*, 2016). All these human outbreaks occurred in rural areas in China, where household pork farming and consumption is common (Rayanakorn *et al*, 2018). *Streptococcus suis* infection in humans manifests as meningitis, streptococcal toxic shock-like syndrome, and septicemia (Chen *et al*, 2007; Lun *et al*, 2007; Ferrando *et al*, 2015), often resulting in fatality. Since the first outbreak in 1998, the incidence of *S. suis* infections has been markedly increasing. More than 1,500 cases of human infections have been reported in 34 countries between 2002 and 2013 (Goyette-Desjardins *et al*, 2014). It is likely that *S. suis* infection is largely underestimated due to its prevalence in rural areas lacking proper diagnosis and medical resources (Huong *et al*, 2014; Bojarska *et al*, 2016), posing an emerging concern in public health worldwide. Taking Poland as an example, it was reported that half of *S. suis* infections were missed initially or misdiagnosed (Bojarska *et al*, 2016).

Our knowledge of *S. suis* pathogenesis and adaptation in humans is highly limited, whereas most population genomic studies to date have focused on strains isolated from healthy and diseased pigs with very few isolates from human. A recent genomic analysis of 98 isolates from Netherlands revealed that the disease-associated *S. suis* isolates have reduced genome size but increased number of putative virulence factors (Willemse *et al*, 2016). This trend was also observed in another analysis of 191 isolates from Vietnam, which established a clade of *S. suis* that are strongly associated with

1 Key Laboratory of Environment Correlative Dietology, Interdisciplinary Sciences Institute, College of Food Science and Technology, Huazhong Agricultural University, Wuhan, China
2 National R&D Center for Se-rich Agricultural Products Processing, Hubei Engineering Research Center for Deep Processing of Green Se-rich Agricultural Products, School of Modern Industry for Selenium Science and Engineering, Wuhan Polytechnic University, Wuhan, China
3 The Center for Microbes, Development and Health (CMDH), CAS Key Laboratory of Molecular Virology and Immunology, Institut Pasteur of Shanghai, Chinese Academy of Sciences, Shanghai, China
4 State Key Laboratory of Agricultural Microbiology, Huazhong Agricultural University, Wuhan, China
5 College of Fisheries, Huazhong Agricultural University, Wuhan, China
6 College of Veterinary Medicine, Nanjing Agricultural University, Nanjing, China
7 Laboratory of Bacterial Pathogenesis and Immunology, The Rockefeller University, New York, NY, USA
8 Institute of Translational Medicine, School of Medicine, Zhejiang University, Hangzhou, China
*Corresponding author. Tel: +86 027 87282111; E-mail: pandafengye@zju.edu.cn
**Corresponding author. Tel: +86 571 88981576; E-mail: lijinquan2007@gmail.com
†These authors contributed equally to this work

invasive disease in pigs (Weinert *et al*, 2015). However, the limited number and geographic diversity of human *S. suis* isolates in previous studies prevented a full understanding of genomic evolution of *S. suis* and their recent jump and pathogenesis in humans.

In this study, we have obtained the genome sequences of 366 *S. suis* clinical isolates, with 103 human isolates collected since most recent outbreaks. Combining with 1,268 available *S. suis* genomes worldwide, we have performed the largest genomic analysis to date and discovered a new clade of human-associated *S. suis* that are distinct from (healthy and diseased) pig isolates. Our analysis has revealed the global dissemination of human-associated *S. suis* from Europe to outbreaks in East Asia. In addition, we have established a set of genetic markers for differential diagnosis and enhanced surveillance in food industry to help prevent future *S. suis* outbreaks in humans.

# Results

## A human-associated clade from the global *Streptococcus suis* population

To understand the genetic relationship of *S. suis* populations from humans and pigs, we have Illumina-sequenced the genomes of 366 clinical strains isolated in China and compared with the genome sequences available in GenBank, culminating into a compendium of 1,634 genomes from 14 countries. Using the well-annotated BM407 genome as reference, we have identified a total of 397,901 SNPs in this collection and analyzed their population structure using Bayesian analysis of population structure (BAPS) (Chen *et al*, 2013; Ruan & Feng, 2016). This analysis has identified nine BAPS groups, each sharing a similar set of SNPs, which is indicative of a common ancestry with similar admixture. Interestingly, 96% of the human isolates fell into the BAPS7 group (Appendix Fig S1A and B), including all the highly virulent isolates of sequence type 1 (ST1) and the epidemic isolates of ST7. Serotypes 2 and ½, the most common serotypes in human infections (Rayanakorn *et al*, 2018), were mostly found in BAPS7 too (Appendix Fig S1C). Therefore, BAPS7 likely represents a dominant group of virulent *S. suis* associated with human infections. This observation is not confounded by potential bias in sampling locations, because the human isolates from different countries were clustered in BAPS7 and swine isolates from one country were dispersed in all nine groups (e.g., China and UK, Appendix Fig S1D).

Next, we have constructed a maximum-likelihood phylogenetic tree for the nine BAPS populations. The topology of the phylogeny shows that the tree is segregated into three main clades, and each clade is highly associated with a distinct host source (Fig 1A). The vast majority of the human isolates (549/570, BAPS7) were clustered together in one clade, which will be referred to as the human-associated clade (HAC). The other two clades are defined as the diseased-pig clade (DPC) or healthy-pig clade (HPC), because the majority of isolates (69%, 181/262) in DPC were obtained from diseased pigs, and most isolates from healthy pigs (72.2%, 279/386) clustered in HPC (Fig 1B). The human isolates in HAC have significantly smaller genome size (2003 ± 47 genes), than the swine isolates in DPC (2,130 ± 63 genes) and HPC (2,191 ± 147 genes) (Fig 2A). Furthermore, the partitioning between three clades was

confirmed by another phylogenetic analysis using the non-recombinant regions (Appendix Fig S2) and by principal component analysis (PCA) based on the core-genome multi-locus sequence typing (cgMLST) scheme (Appendix Fig S3), bolstering the identification of a novel human-associated clade.

To understand whether these human-associated strains possess high virulence potential, we have randomly selected 14 isolates in HAC and tested their pathogenesis in an established infection model for *S. suis* (Neely *et al*, 2002; Zaccaria *et al*, 2016). In addition, we have tested ten healthy-pig isolates from HPC and included the SC19 strain responsible for the 2005 human outbreak as a positive control. As depicted in Fig 2B, all 14 isolates from HAC displayed a significantly higher mortality than the HPC isolates (*P* < 0.0001, Fisher's exact test). As expected, the positive control SC19 caused a high degree of mortality in this model of infection (Appendix Table S1), very similar to the new HAC strains tested. Notably, 5 of these HAC strains were originally isolated from healthy pigs, indicating that healthy pigs may be a reservoir of HAC strains with high virulence potential in human.

## Occurrence and divergence of China-specific ST7 lineages since the 1960s

Further inspection of HAC has identified three sub-lineages (I, II, and III) (Dataset EV1, Appendix Fig S4). Lineages I and III are dominated by the China-specific sequence type ST7, which is considered as a highly virulent variant (Zhou *et al*, 2017). Notably, lineage I is responsible for the first two large-scale human outbreaks in China, in 1998 (Jiangsu) and 2005 (Sichuan). According to our time-dated phylogeny constructed using 174 representative isolates (selected based on sampling time and geographic diversity, see below and methods), the outbreak isolates from Jiangsu and Sichuan diverged in 1993 (95% CI, 1991–1995) and 1996 (95% CI, 1993–1998), respectively, consistent with historical reports (Ye *et al*, 2006; Du *et al*, 2017). Genomic analysis also confirmed that the lineage I strains possess an 89-kb pathogenicity island (PAI) (Fig 3B, Appendix Fig S4).

Lineage III represents a novel ST7 type. It mostly likely emerged in the UK around 1969 (95% CI, 1963–1977). Lineage III strains lack the 89-kb PAI, but encode numerous antibiotic resistance genes including *tet*(O), *tet*(40), *mef*(A), *msr*(D), *erm*(B), *aph*(3′)-*III*, *sat4*, and *ant(6)-Ia*. Resistance to tetracycline, clindamycin, and erythromycin was confirmed for all 19 ST7 isolates encoding these antibiotic resistance genes in our collection (Dataset EV1). Furthermore, we have identified 69 genes that are specific to lineage III (Appendix Table S2), located on two large regions. The first region contains 17 genes, and the second region contains 52 genes located within a putative 127-kb mobile genetic element (MGE). This MGE encodes all eight antibiotic resistance genes described above and appears to be a long tandem MGE containing a prophage–ICE or ICE–prophage flanked by three *att* sites (Fig 4A). The prophage was inserted at the 3′ end of an RNA uracil methyltransferase (*rum*) gene, which is a conserved insertion hotspot for MGE integration in *S. suis* and other streptococci (Palmieri *et al*, 2011).

To determine whether the 127-kb MGE is transmissible, we designed primers to specifically amplify the excised form and confirmed that both ICE and prophage indeed excise from the chromosome (Appendix Fig S5). Since all *S. suis* strains have *rum* locus and *att* site, it is likely that this 127-kb island and the encoded drug-

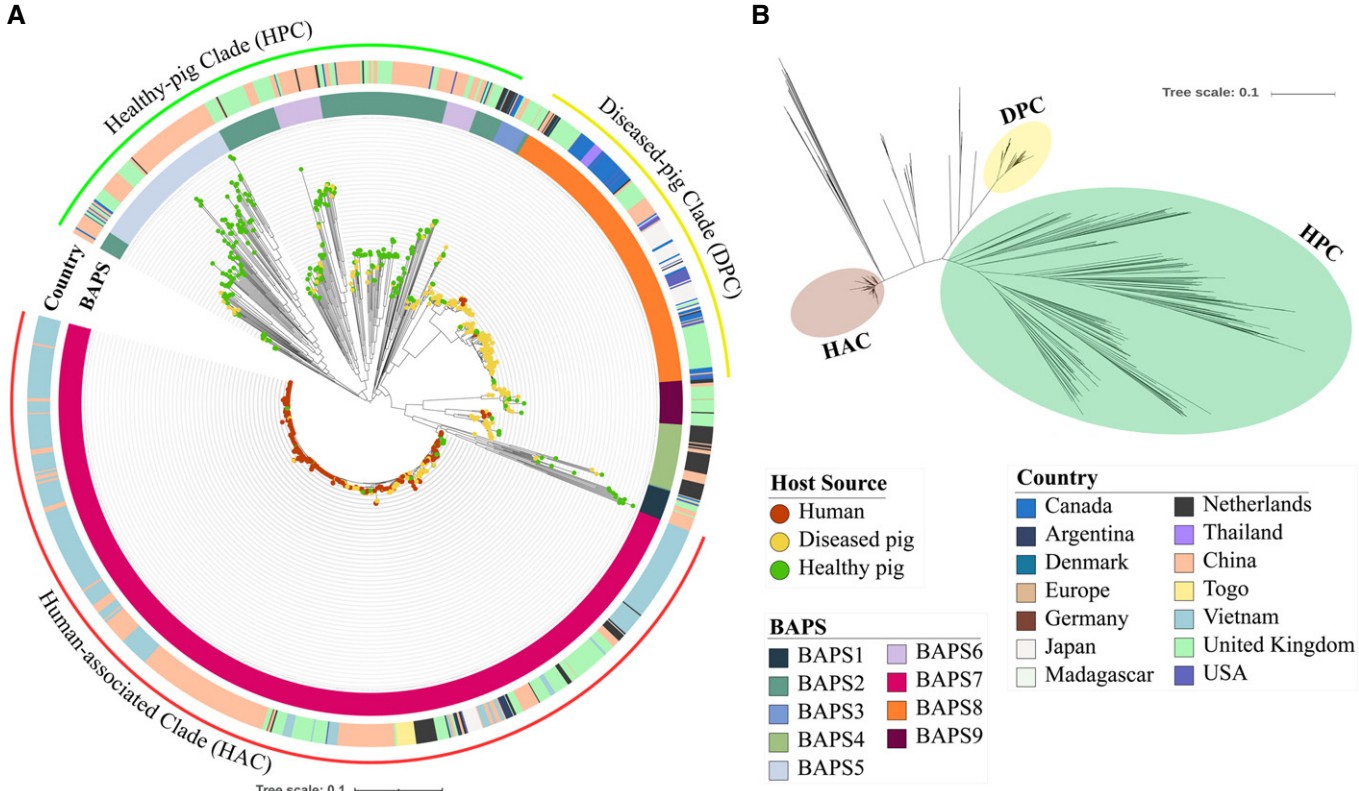

**Figure 1.   Identification of the human-associated clade from *Streptococcus suis* population.**

A   Correlation of the maximum-likelihood tree with genetic and phenotypic classification/clustering. Maximum-likelihood tree was constructed using genome-wide SNPs of 1,634 *S. suis* isolates. Tip nodes are colored based on the host source (human in red; diseased pig in yellow; and healthy pig in green). From inside to outside: The first ring represents BAPS clustering; the second ring represents the country source. Human-associated clade (HAC) are colored in red, with diseased-pig clade (DPC) in yellow and healthy-pig clade (HPC) in green. Phylogenetic relationship is consistent with BAPS clustering and correlates with host source. Most isolates in BAPS2, BAPS5, and BAPS6 were obtained from healthy pigs. The majority of isolates in BAPS7 and BAPS8 were obtained from patients and diseased pigs, respectively.

B   Phylogenetic tree (unrooted) showing three important clades (HAC in red; DPC in yellow; and HPC in green).

resistant genes can horizontally transfer to other strains. In fact, we have found this MGE in 13 human isolates from Guangdong, Guangxi, and Zhejiang provinces, as well as ten pig isolates from Sichuan, Jiangsu, Hunan, Hubei, and Guangxi provinces, indicating the widespread of MGE among multi-drug-resistant ST7 isolates in China (Appendix Fig S6).

### Emergence and circulation of the epidemic ST1 lineage between China and Vietnam

Phylogenetic analysis demonstrated that 58.8% (113/192) isolates from Vietnam clustered closely with isolates from China, forming a new Asian lineage (lineage II, Appendix Fig S4). The emergence of lineage II was dated to 1966 (95% CI, 1959–1973), based on our time-dated phylogeny analysis (Fig 3). Within this lineage, 39 human isolates in our collection were from the Guangxi Province of China, which shares border with Vietnam. Clinical information argues for the high virulence potential of lineage II. One strain was isolated from a person who died of a severe infection after eating "high-risk" pork. Another strain was isolated from a patient diagnosed with toxic shock-like syndrome who died 2 days after starting treatment, during the most recent outbreak in China in 2016 (Huang

*et al*, 2019). Within lineage II, ST1 is the most common strain type (96%, 154/161) responsible for human infections, consistent with previous report that ST1 was mostly associated with disease in both humans and pigs in Asia (Segura *et al*, 2017). ST1 strains are also responsible for *S. suis* disease in humans worldwide (Fittipaldi *et al*, 2011; Goyette-Desjardins *et al*, 2014), raising concerns about this emerging epidemic lineage.

Analysis of accessory genomes has identified 46 genes that are specific to lineage II (Appendix Table S3), 39 of which are located on a novel 78-kb pathogenicity island (PAI). This PAI contains SalKR, NisKR, a type IV-like secretion system, and a Tn916 element, all of which are essential for the full virulence of strains that were involved in the recent outbreaks (Fig 4B).

Besides ST1, the 78-kb PAI is also found in several other STs (ST107, ST658, ST869, ST951, ST1005). The excision of 78-kb PAI from ST1 genomes indicates a high probability of its spread around the world. Molecular clock analysis revealed that lineage II diverged at a different time point compared to lineage I. Thus, the 2005 outbreak lineage I (causing 2005 outbreak) and lineage II (causing 2016 outbreak) may have appeared through independent routes of molecular evolution after acquiring the 78-kb and 89-kb PAI, respectively.

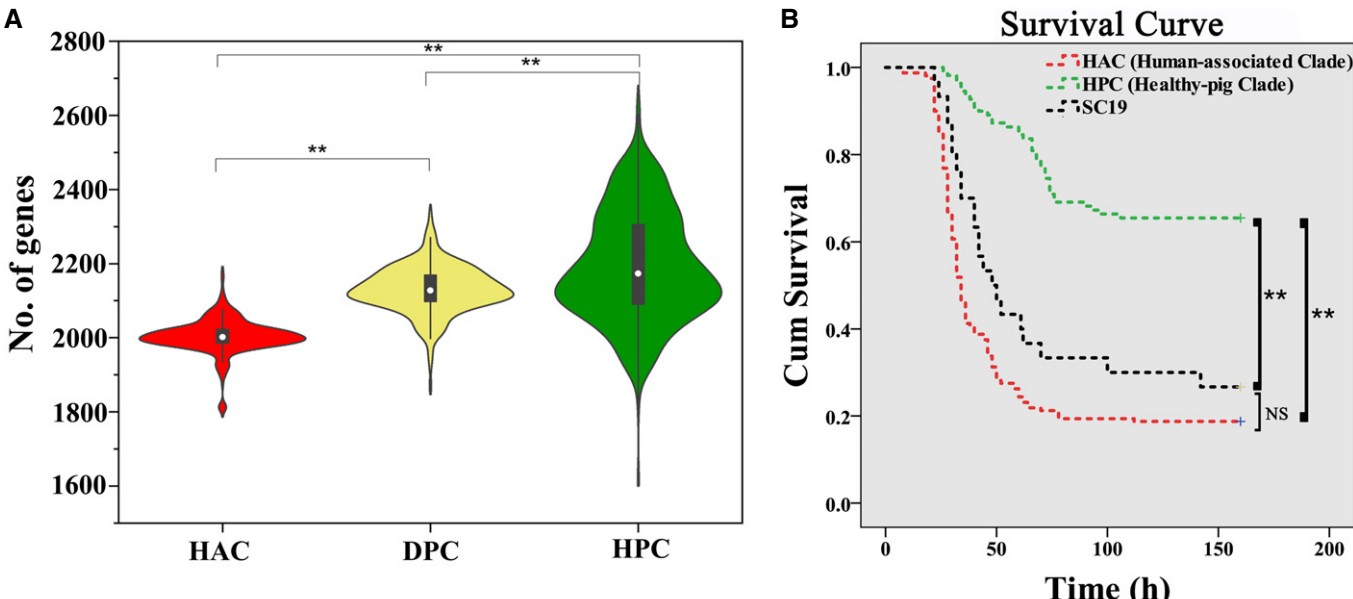

**Figure 2. Isolates from human-associated clade (HAC) are more virulent than isolates from healthy-pig clade (HPC).**

A   The violin plot depicting the smaller genome size of HAC isolates ($n = 820$) than that of DPC ($n = 407$) and HPC ($n = 262$). Inside the violins are the box and whisker plots, with the dot showing the median, the box showing the quartiles, and the whisker showing the 95% percentiles. *P*-values were calculated using unpaired *t*-test (**$P < 0.05$).

B   The survival curve for zebrafish inoculated with representative *S. suis* isolates from HAC and HPC.

Data information: In total, twenty-five representative isolates were tested, including fourteen isolates from HAC and ten healthy-pig isolates from HPC, as well as the strain SC19 in HAC responsible for the 2005 human outbreak as a positive control. The cumulative mortalities were calculated using the combined data from the isolates in each group. As expected, all 14 isolates from HAC displayed a significantly higher mortality than the HPC isolates ($P < 0.05$). Survival curve and statistical analysis were created using Kaplan–Meier and SPSS 23.0. *P*-values were calculated using log-rank test. $P < 0.05$ was considered significant (**$P < 0.05$; NS, $P > 0.05$). No significant difference (NS, $P = 0.072$) was found between HAC isolates and SC19.

## Transcontinental dissemination of the human-associated clade from Western Europe in the 19th Century

To better understand the temporal emergence of HAC, we have constructed a time-dated phylogeny and estimated a timeline for the intercontinental and regional spread by analyzing 174 representative HAC isolates from different countries, including Argentina, China, Japan, Madagascar, Netherlands, Togo, United Kingdom, United States, and Vietnam (Dataset EV1, Supplementary Methods). The most recent common ancestor of HAC was dated back to 1830 (95% credible interval (CI), 1802–1855) and likely emerged in Western Europe. Thereafter, the human-associated strains migrated to other major continents (Africa, America, and Asia) and resulted in a stable establishment in Asia (Fig 5), the continent with the highest prevalence of *S. suis* presently (Huong *et al*, 2014). These introduced *S. suis* strains in Asia are the direct cause of the first two major human outbreaks in China in 1998 and 2005 (Fig 5). By reconstruction of the demographic history using the Bayesian skyline plot, we revealed that the population of human-associated *S. suis* experienced multiple shifts in size. The population size was relatively constant from 1836 to 1952 and then expanded and spread to other continents between 1954 and 1988 (Fig 3A), coinciding with the global dissemination of advanced pig breeds from European breeders throughout the 1960s and 1970s (Callejo *et al*, 2016).

## Clade-specific signatures contribute to the virulence of human-associated isolates

To further explore accessory genes that may be associated with the evolutionary success of the human-associated strains, we have performed GWAS analysis and chi-square analysis to identify genes that are specific to HAC. While GWAS analysis revealed 66 HAC-specific genes (Appendix Fig S7, Appendix Table S4), chi-square test identified a core set of 25 genes, all of which were also found by the GWAS analysis (Appendix Fig S8, Appendix Table S5). Of these 25, nine genes are located in the same region, which is not identified as MGE according to VRprofile, Phage_Finder, and ICEberg database. The other 16 genes are distributed at different locations in the genome (Appendix Fig S9). Most genes (18/25) encode hypothetical proteins or metabolic enzymes. Of note, three genes have been identified as putative virulence factors (*pnuC*, *epf*, and *nadR*). PnuC is an importer for nicotinamide riboside in the nicotinamide salvage pathway. It has been shown to be involved in the oxidative stress tolerance and virulence of *S. suis* during infection (Wilson *et al*, 2007; Li *et al*, 2018b) and also contributes to virulence of *Streptococcus pneumoniae* (Johnson *et al*, 2015). Epf was considered as a key virulence marker for *S. suis* (Goyette-Desjardins *et al*, 2014). *Streptococcus suis* strains with Epf have been isolated from blood and cerebrospinal fluid in human patients in Thailand and are considered

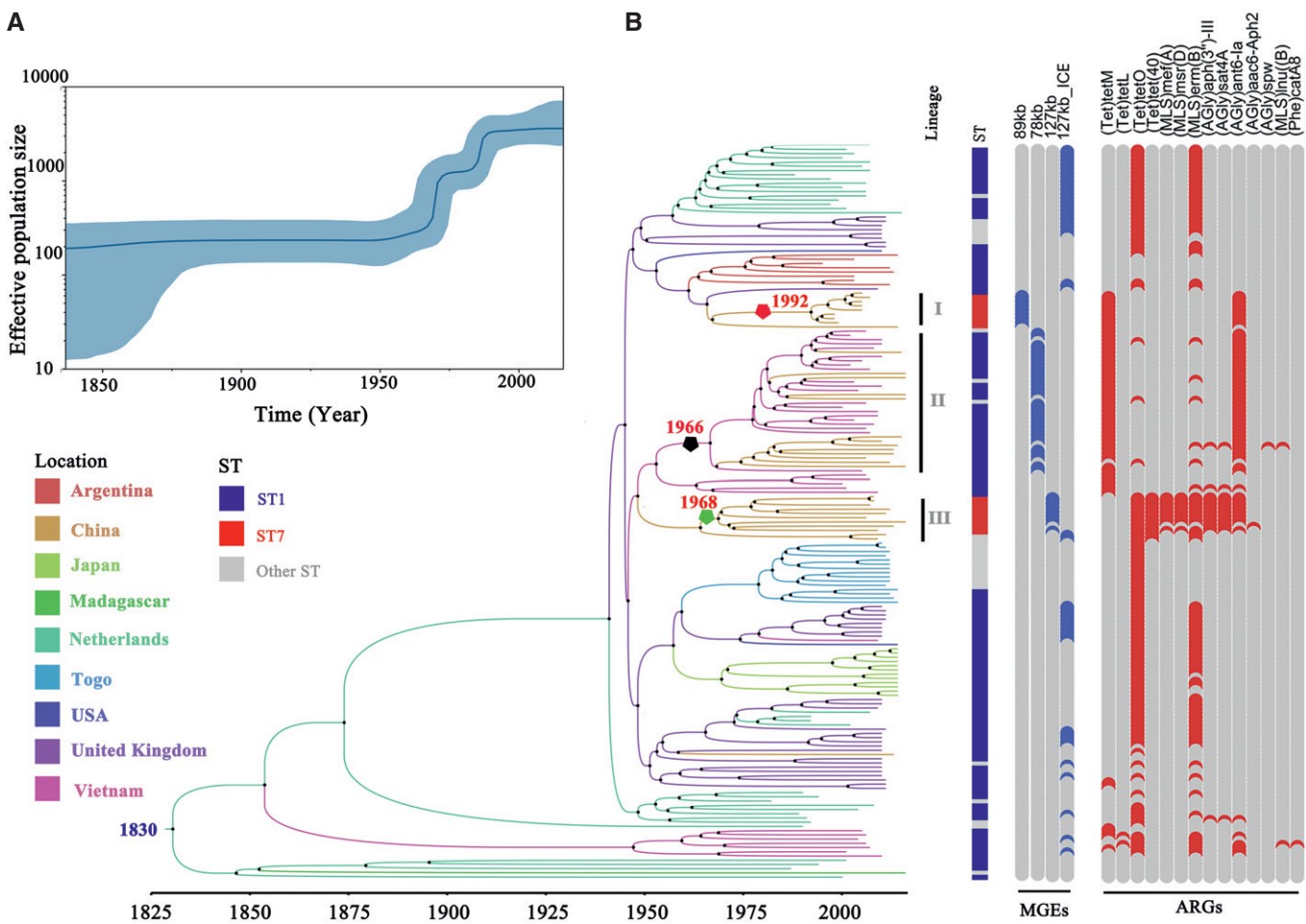

**Figure 3. Bayesian skyline plot (BSP) and phylogenetic tree of human-associated *S. suis*.**

A   Bayesian skyline plot showing the population size through time for human-associated *S. suis*, which is constructed based on a subsample of HAC isolates (*n* = 174). The y-axis represents the effective population size, and the x-axis is calendar years. The line shows the median estimate of the population size. Blue shading shows 95% highest posterior density.

B   Timed phylogeny of the subsample of HAC isolates (*n* = 174). Maximum clade credibility tree is produced using strict-clock model in BEAST2. Major sub-lineages in Asia are indicated (I, II, and III). Acquisition of the 89-kb pathogenicity island is indicated by a red pentagon. Acquisition of the 78-kb pathogenicity island and the 127-kb MGE is indicated by black and green pentagon, respectively. The blue and red blocks of heatmaps represent the presence of MGEs and ARGs, respectively. Gray represents absence.

highly virulent (Kerdsin *et al*, 2011). NadR was reported as a transcription factor modulating *S. suis* pathogenesis in pigs (Okura *et al*, 2017). It has been found in a number of highly pathogenic strains (Zheng *et al*, 2011; Huang *et al*, 2019), as well as recent ST7 isolates (the sequencing type responsible for two human outbreaks in China) (Wang *et al*, 2019). Therefore, these HAC-specific genes might directly contribute to the increased risk of human infection.

Our further analysis on whole-genome SNPs has revealed 71 SNPs that were significantly associated with human-associated isolates (Appendix Fig S10A). All SNPs were found within protein-coding regions, resulting in 17 (24%) missense variants and 54 (76%) synonymous variants. The missense SNPs affected 16 genes of various functions (Appendix Table S6), similar to the results of COG analysis for accessory genes (Appendix Fig S10B). Intriguingly, we have identified ten genes that are specific to DPC (Appendix Fig

S8, Appendix Table S7). This provides strong evidence to support the separation of DPC from pig isolates in HAC, which may have diversified into human-associated strains.

Using these marker genes, we have successfully developed a differential diagnosis method that can identify human-associated strains that are capable of causing disease in human. Receiver operating characteristic (ROC) curve analysis demonstrated that these 25 genes have good predictive power, with the area under the curve (AUC) value > 0.9 (Appendix Fig S11). Validating two selected marker genes using in silico PCR showed that both primer pairs efficiently amplified the target regions in HAC strains (sensitivity > 99% and specificity > 99%) (Dataset EV2). The expected amplicons of 1,065 and 1,965 bp were produced by all 12 training isolates in HAC (Appendix Fig S12) in PCR amplification. Another 21 previously uncharacterized patient isolates were also positive using these primers (Appendix Table S8). Other published isolates with high

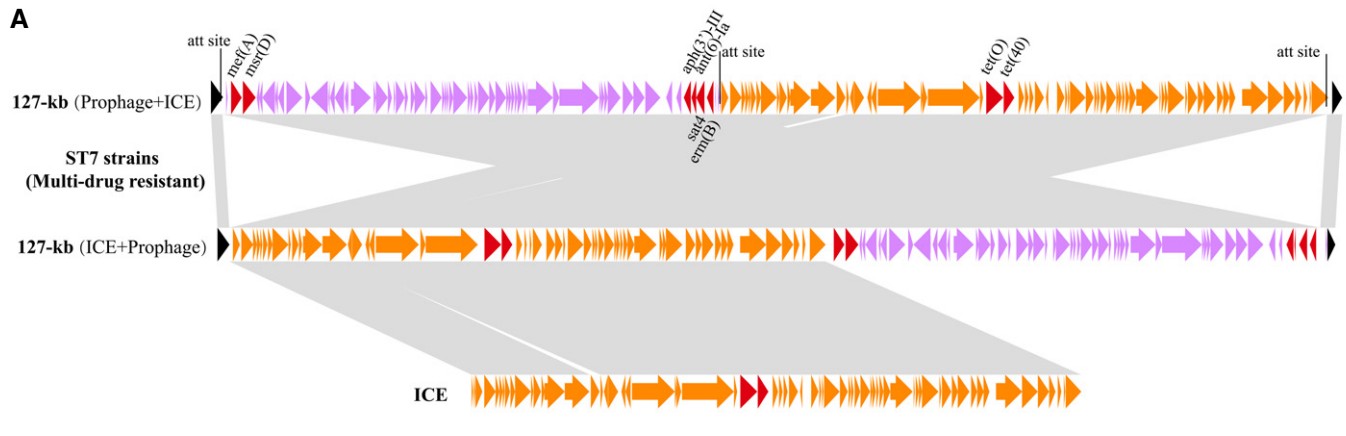

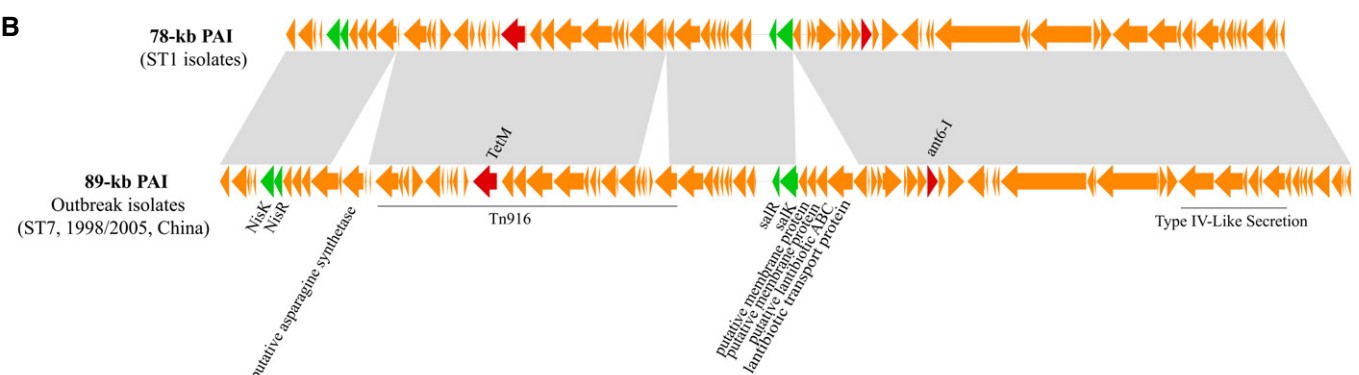

**Figure 4.  Schematic diagrams of 127-kb MGE and 78-kb PAI derived from lineages III and II, respectively.**

A    Two types of 127-kb tandem MGE detected in multi-drug-resistant *S. suis* ST7 isolates in lineage III. The 127-kb MGE encodes eight antibiotic resistance genes. The integrative and conjugative elements (ICE, in orange) and prophage (in purple), both of which are flanked by att sites, are tandemly arranged in two possible orders. The antibiotic resistance genes are indicated in red. The chromosomal conserved genes around MGE are indicated in black.

B    Comparison of the novel 78-kb PAI reported in this study, with the 89-kb PAI reported in epidemic strains from two outbreaks in China. The key factors for virulence of epidemic strains are indicated in green, including SalKR, NisKR, a type IV-like secretion system, and a Tn916 element. The antibiotic resistance genes are shown in red. Three regions in the 89-kb PAI are absent from the 78-kb PAI. No virulence-related genes were found within these regions.

virulence and human infecting potential (Yu *et al*, 2016; Qian *et al*, 2018) were also tested positive. As an important control, all 10 low-virulence isolates from pigs were tested negative (Appendix Fig S13). Therefore, these HAC-specific genes could serve as valuable markers for the development of DNA-based diagnostics, for detection and surveillance of human-associated *S. suis* strains that have high virulence potential of causing disease in humans.

## Discussion

*Streptococcus suis* is increasingly recognized as a preventable emerging zoonotic infection in humans with a global distribution (Rajahram *et al*, 2017). The number of *S. suis* cases has notably increased during the past few decades, with the highest prevalence rate in rural Asia (Rayanakorn *et al*, 2018). Three serious outbreaks in human have been documented in China in 1998, 2005, and most recently in 2016 in the Guangxi Province (Tang *et al*, 2006; Huang

*et al*, 2019). The 2005 outbreak affected more than 200 people with a mortality rate of nearly 20% (Gottschalk *et al*, 2010). These accumulating incidences argue that certain porcine strains of *S. suis* may be evolving to "high-risk" human pathogens. In this study, we have identified a novel clade strongly associated with human infections (HAC) and revealed its genetic markers and evolutionary history. Our data showed that HAC emerged in the 1830s in Western Europe and subsequently transmitted to the rest of the world, confirming a previous hypothesis that the zoonotic isolates in Asia likely originated from United States or European Union (Weinert *et al*, 2015).

Using a large collection of strains from 14 countries, our phylogeographic analysis provided new insight into the global population structure of *S. suis*. We have shown a clear separation of strain clades associated with different host source. However, a previous analysis of *S. suis* isolates did not report any genomic difference or splitting between human and pig hosts. This may be due to the limited number of human isolates analyzed or that these human isolates were restricted to a single source (Vietnam). Based on these

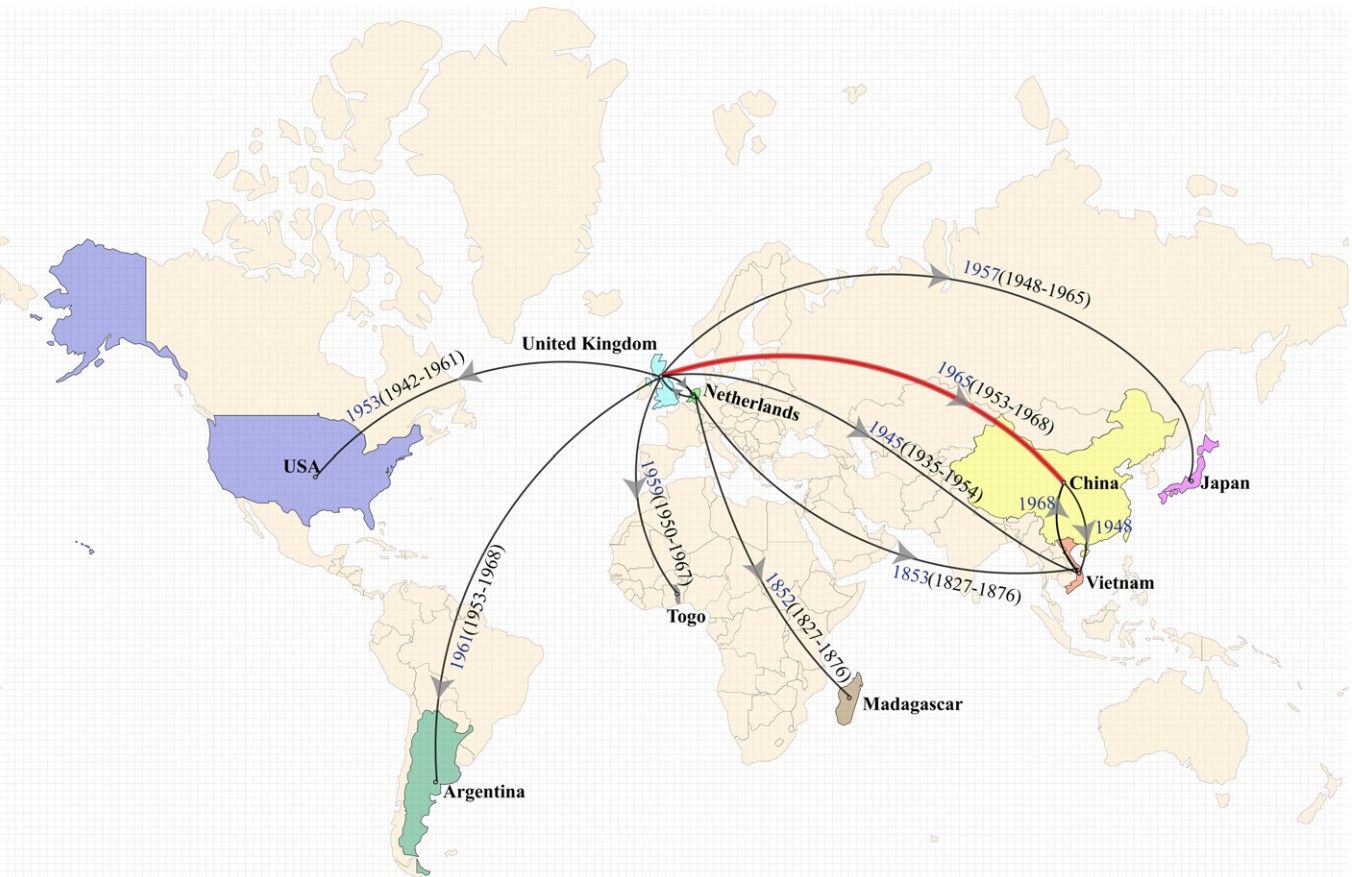

**Figure 5. Phylogeographic analysis revealed global transmission patterns of human-associated *S. suis*.**

Countries of relevance in the map are colored. Geographic presence, inferred arrivals (arrows), and principal long-distance transmission events (line) are based on phylogeographic analysis. The data are the inferred values for the most recent common ancestor (taken from BEAST2). The origin of most transmission waves was Europe. One of these transmission waves was spread to China leading to two major outbreaks (Jiangsu and Sichuan) (red line).

published Vietnamese isolates and new human isolates in our study, we have for the first time identified three distinct clades for *S. suis*: HAC, DPC, and HPC. Whereas the HAC and DPC strains formed tight clusters, the HPC isolates from healthy pigs have longer phylogenetic branches (Fig 1B) and a larger pan genome without obvious clade-specific genes (Appendix Fig S14), reflecting their great genetic diversity. This adds support to the hypothesis that healthy pigs (i.e., carrier pigs) may be the early reservoir of *S. suis* and the source of human-associated strains.

Intriguingly, our analysis suggests that the *S. suis* isolates involved in the recent outbreak (2016) in China may be closely related with Vietnamese isolates. Appeared almost in parallel with isolates from the two outbreaks, the lineage II of HAC is an epidemic lineage currently spreading across the borders of Vietnam and China. International collaboration and commitments will be required for the effective surveillance of this lineage. Most of linage II epidemic strains are ST1, one of the most prevalent and virulent clones worldwide. These ST1 isolates often contain the 78-kb PAI, in stark contrast to other outbreak strains in China that are restricted to ST7 (the China-specific ST).

Considering the high mortality of *S. suis* infections in humans, it is critical to properly surveil and detect potential HAC strains in

pork production chain and clinics. Our comparative genomic analyses have identified 25 genes that are highly specific to HAC, whereas previous studies have reported a different set of genes associated with DPC (Weinert *et al*, 2015; Guo *et al*, 2020). Based on these genes, we have developed a novel diagnosis method to rapidly distinguish the virulent HAC isolates from Asia. Furthermore, since 40% of these marker genes encode membrane associated and secreted proteins (Dataset EV3), these antigens may be considered as potential candidates for vaccine development in the future.

# Materials and Methods

### Bacterial collection

144 *S. suis* strains (Dataset EV4) were isolated from 2005 to 2016, including 103 isolates from human patients and 41 from diseased pigs. 222 strains were isolated from healthy pigs in a previous study by our group (Zou *et al*, 2018). Isolates and clinical information were obtained from hospitals and veterinary institutes. 1,490 (excluding our isolates) whole-genome sequences of *S. suis* with

known country of origin and host source were obtained from GenBank (downloaded in March 21, 2018, Dataset EV4).

## Antimicrobial susceptibility testing

In vitro antimicrobial susceptibility testing was performed on Mueller–Hinton (MH) agar by the Kirby–Bauer disk diffusion method (Tran-Dien *et al*, 2018). The culture medium was supplemented with 5% defibrinated sheep blood according to the standardized methods described by the Clinical and Laboratory Standards Institute (CLSI, 2015, 2016). The antimicrobial agents used in this study were penicillin G (1 unit), cefuroxime (5 μg), tetracycline (30 μg), chloramphenicol (10 μg), vancomycin (30 μg), gentamicin (30 μg), erythromycin (15 μg), clindamycin (2 μg), rifampicin (5 μg), imipenem (10 μg), and levofloxacin (1 μg). The method used was in accordance with the CLSI. Briefly, three or four colonies from an overnight culture on Columbia agar supplemented with 5% sheep blood (AES) were suspended in MH broth. The suspension was adjusted to a 0.5 McFarland standard and diluted to obtain an inoculum of CFU/ml *S. suis*. For each strain, two plates were inoculated by flooding MH agar (4 mm depth) supplemented with 5% defibrinated sheep blood. Antibiotic disks were placed with a disk dispenser (Oxoid, England), and plates were incubated at 37°C in 5% $CO_2$ for 20–24 h. The diameter of the inhibition zone of growth of *S. suis* strains and reference strains was measured on the same day using a sliding caliper. *Streptococcus pneumoniae* ATCC 49619 was used as the quality control strain. CLSI guidelines were used for interpretation of zones of inhibition.

## Genome sequencing

Genomic DNA was prepared from isolates grown overnight at 37°C in Tryptone soy broth (TSB, BD Biosciences) plus 10% bovine serum, using Bacterial DNA kits (OMEGA Bio-Tek). The isolated DNA was sequenced using an Illumina HiSeq 2500 (Novogene, China) with 2 × 150-bp paired-end chemistry, according to the manufacturer's instructions, with > 500 average coverage. Raw data were processed in four steps, including removing reads with 5 bp of ambiguous bases, removing reads with 20 bp of low quality (≤ Q20) bases, removing adapter contamination, and removing duplicate reads. The de novo assembly was performed using SPAdes 3.5.0 (Bankevich *et al*, 2012). The genome assembly statistics were summarized in Dataset EV5. Assembled genomes were uploaded to the online bioinformatics tools SpeciesFinder v2.0 (Larsen *et al*, 2014) available from the Center for Genomic Epidemiology (CGE, Technical University of Denmark, Lyngby, Denmark) services to confirm species. To avoid artificial differences resulting from different annotation pipelines, the assemblies as well as the genomes downloaded from GenBank were reannotated with PROKKA 1.7 (Seemann, 2014).

## Construction of phylogeny

We first aligned the contigs of each of 1,634 *S. suis* sequences against the reference genome (BM407) using MUMmer v3.1 (Delcher *et al*, 2003) to obtain all potential SNP loci. Indels and adjacent mismatches are not considered as true SNPs (Ruan & Feng, 2016). A total of 397,901 sites were polymorphic, and the

accordingly concatenated SNPs were used for building a maximum-likelihood tree by RAxML v8.1.24 software with GTRGAMMA model (Stamatakis, 2014). Recombinant regions were filtered from the alignment and a maximum-likelihood phylogenetic tree generated using Gubbins v2.4.0 (Croucher *et al*, 2015). The obtained phylogenetic tree was visualized with iTOL version 3.5.3 (Letunic & Bork, 2016). A tree-independent hierarchical Bayesian clustering with hierBAPS v6.0 (Cheng *et al*, 2013) was performed to determine of the population structure using hierBAPS. Within a Bayesian framework, the approach first determines the optimal number of genetically distinct clusters (populations) (K) such that the genetic variation within clusters is minimized and the variation among them is maximized. The concatenated SNPs were used as the input; three levels of clustering were performed in the hierarchy; and a prior upper boundary of 20 clusters was set. The estimated number of clusters was 9 for levels 1, 2, and 3. Population assignment coefficients were obtained using 100 iterations. The level of admixture (mixed ancestry) was obtained using 10 reference individuals and 50 iterations.

To determine the population structure of HAC, a maximum clade credibility tree was constructed based on concatenated SNPs of 562 genome sequences (Dataset EV1) that have attached sampling dates in HAC. The time of the most recent common ancestor (tMRCA, years) and spatiotemporal dynamics of dissemination of HAC isolates was estimated with BEAST v2.5 (Bouckaert *et al*, 2019), with the HKY85 plus Gamma nucleotide substitution model, Bayesian Skyline coalescent tree prior, and a strict clock. To ensure that there was no sampling bias, 174 representative isolates were selected for the study based on sampling time and geographic location. The analysis was performed with the SNPs from the non-recombinant regions of 174 *S. suis* whole-genome alignment as matrix. During these analyses, a total of 100,000,000 MCMC chains were explored every 10,000 steps, and the first 10% samples were discarded as burn-in. The estimation of the relevant evolutionary parameters was checked using Tracer v1.7.1. The tree was produced using TreeAnnotator v1.8.4 and displayed in FigTree v.1.4.3 (Rambaut, 2014; http://tree.bio.ed.ac.uk/software/figtree). The phylogeographic inference was analyzed and visualized with the spatial phylogenetic reconstruction of evolutionary dynamics using data-driven documents (SpreaD3) v 0.9.659 (Bielejec *et al*, 2016).

## Molecular serotyping and multi-locus sequence typing (MLST)

A BLAST database was built based on a published multiple PCR method (Okura *et al*, 2014), which included all unique cps genes for *S. suis* serotype identification. Sequence identities > 95% with an alignment length > 95% of the target gene were used as the threshold for prediction of gene presence/absence. The cps genes of serotypes 2 and 1/2 are too similar to distinguish at the draft genomic level; thus, the two serotypes were grouped together. Likewise, serotypes 1 and 14 were also grouped together. The non-serotypeable isolates were further classified into 8 different groups according to 8 novel cps loci (NCL) (Zou *et al*, 2018). The MLST analysis was performed using the online service BacWGSTdb (Ruan & Feng, 2016). MLST alleles and sequence types (STs) were assigned through the submission of the respective genome data to the *S. suis* PubMLST database (https://pubmlst.org/). The cgMLST (Core Genome Multilocus Sequence Typing) analysis was performed

using the online service BacWGSTdb (Ruan & Feng, 2016). The obtained allele matrix was used for performing a principal component analysis (PCA) by using the FactoMineR package.

**Pan-genome analysis of potential markers**

Based on annotated assemblies, a pan genome was calculated for all 1,634 isolates using Roary (Page *et al*, 2015). A total of 27,524 non-redundant coding sequences (CDS) were identified, 27,066 of which were assigned to the accessory (variably present) genome. We compared the accessory genes in isolates from different groups (host-related) and identified the respective host-specific genomic regions. Accessory genes leading to a *P*-value $< e^{-30}$ sorted by significance after performing a chi-square test among three clades. Significant genes were further defined as group special genes (markers) if they occur in one group at > 95% frequency ("consensus"), but < 5% frequency in other groups. We also performed GWAS analysis with GEMMA software (v0.98.1) to search for accessory genes and SNPs that are tightly associated with the disease phenotype of the isolates namely HAC and non-HAC. The *P*-value higher than the genome-wide suggestive thresholds was identified as significant genes or SNPs. Gene function was annotated based on the eggnog database (clusters of orthologous groups). The sub-cellular localization of gene encoded products was predicted with LocTree3 (Goldberg *et al*, 2014). Furthermore, we designed primer sets targeting parts of identified regions and conducted specificity testing on the designed primer sets (we defined primer target nucleotide as markers). To evaluate the predictive accuracy of markers, receiver operating characteristic (ROC) curve analysis was used (Hajian-Tilaki, 2013). In order to confirm if the primer pairs work well in silico, they were tested using a perl script publicly available at https://github.com/egonozer/in_silico_pcr. To further assess the validity of these markers, two groups of *S. suis* isolates were used in this study: (i) a training collection of 24 isolates (12 isolates from HAC, 6 each from clade B1 and clade B2, respectively) and (ii) an out-of-sample test collection of 43 isolates (21 previously uncharacterized isolates from patient and 22 isolates (21 isolates from pigs and an isolate from patient) with known virulence published before) (Yu *et al*, 2016; Qian *et al*, 2018). PCR was performed for 5 min at 95°C, followed by 30 cycles of 30 s at 95°C, 30 s at 55°C, and 2 min at 72°C.

**Identification of mobile genetic elements**

The accessory genomes of *S. suis* isolates were used for genomic comparisons among different clades or lineages in order to identify MGE candidates unique to either one of the groups. MGE candidates were roughly located when genomes were compared with *S. suis* reference strain SC19 (CP020863.1) by MAUVE v2.4.0 (Darling *et al*, 2004). MGEs were predicted with VRprofile v2.0 (Li *et al*, 2018a). Prophages were predicted with Phage_Finder v2.1 (Fouts, 2006). ICEs were predicted with and ICEberg v2.0 (Liu *et al*, 2019). Boundaries and insertion sites of both prophages and ICEs were manually checked. Putative insertion sites and *att* sequences were manually identified.

*Streptococcus suis* genome sequences were mapped against the pathogenicity island or mobile genetic element using a pipeline based on the Mummer v3.1 package (Delcher *et al*, 2003). The results were

**The paper explained**

**Problem**

The Gram-positive bacterium *Streptococcus suis* is an emerging zoonotic agent causing fatal infections in human including septicemia and meningitis. In the last several decades, the number of reported human cases of *S. suis* infections has rapidly increased, and three epidemics were recorded in China in 1998, 2005, and most recently in 2016. The recent increase in the global spread of virulent *S. suis* indicates its adaptive evolution in humans. However, this remained largely elusive as no evidence of genetic adaptation to humans was reported.

**Results**

This study reports for the first time a novel clade of *S. suis* that is strongly associated with human infections. This human-associated clade (HAC) has been forming during the global transmission of *S. suis*, likely originated from Europe and subsequently spread to other continents. HAC strains showed high virulence and caused high mortality in an animal model of infection, similar to the isolate responsible for human outbreaks. Several strains isolated from healthy pigs were found in HAC and possess high virulence potential, suggesting that healthy pigs (i.e., pigs carrying *S. suis*) may be an early reservoir of *S. suis* and a significant source for human infection. Our genome-wide analyses have identified discriminative markers for the human-associated clade. These marker genes are key to the development of DNA-based diagnostics.

**Impact**

Our findings provide genetic evidences to understand the emergence of *S. suis* infections in humans in recent years. The identification of human-associated clade necessitates enhanced *S. suis* surveillance in environments and healthcare system. Diagnostic marker genes identified will be instrumental for future surveillance of human-associated strains with high virulence potential.

translated into heat maps representing the presence and absence of the locus in all the isolates using in-house R script. Antimicrobial resistance genes were searched using the BLASTN program and the ResFinder database (https://cge.cbs.dtu.dk/services/ResFinder/).

**Testing virulence of isolates in the zebrafish model**

The virulence of twenty-five representative isolates was assessed in a zebrafish infection model (Neely *et al*, 2002). We have randomly selected fourteen isolates in HAC and ten healthy-pig isolates from HPC. The strain SC19 in HAC was responsible for the 2005 human outbreak as a positive control. In brief, all strains were grown in Tryptic soy broth (TSB) and harvested by centrifugation at the mid-exponential phase ($OD_{600} = 0.6$). Zebrafish were anesthetized with tricaine methanesulfonate (MS-222, Syndel) at a concentration of 90 mg/l and injected intraperitoneally with bacterial suspensions containing $5 \times 10^7$ CFU/zebrafish of one strain. Mortality was monitored every 2 h for 1 week post-infection. Zebrafish survival data of representative isolates in each clade were combined for analysis. Survival curves and statistical analysis were created by SPSS 17.0, and Kaplan–Meier was used to analyze survival curves. $P < 0.05$ was considered significant. Zebrafish experiments were performed according to experimental protocols approved by the Ethics Review Committee for Animal Experimentation of Huazhong Agricultural University, China.

## Data availability

The genome sequences produced in this study are available in the following datasets:

i      https://www.ncbi.nlm.nih.gov/bioproject/PRJNA566245
ii     https://www.ncbi.nlm.nih.gov/bioproject/PRJNA437405
iii    https://www.ncbi.nlm.nih.gov/bioproject/PRJNA437683
iv     https://www.ncbi.nlm.nih.gov/bioproject/PRJNA437689
v      https://www.ncbi.nlm.nih.gov/bioproject/PRJNA437690
vi     https://www.ncbi.nlm.nih.gov/bioproject/PRJNA437693
vii    https://www.ncbi.nlm.nih.gov/bioproject/PRJNA437694
viii   https://www.ncbi.nlm.nih.gov/bioproject/PRJNA437696
ix     https://www.ncbi.nlm.nih.gov/bioproject/PRJNA437698
x      https://www.ncbi.nlm.nih.gov/bioproject/PRJNA427455
xi     https://www.ncbi.nlm.nih.gov/bioproject/PRJNA427457
xii    https://www.ncbi.nlm.nih.gov/bioproject/PRJNA427459

**Expanded View** for this article is available online.

## Acknowledgements

This work was supported by The National Key Research and Development Program of China (2017YFD0500201, 2017YFC1600100), the National Natural Science Foundation of China (32072323, 31772083), Special fund for Technology Innovation of Hubei Province (2019AHB074), the Fundamental Research Funds for the Central Universities (2662017JC040), and China Scholarship Council (201806760032, 201806765004).

## Author contributions

JL and YF designed the study. XD performed experiments. XD and YF performed genomic data analysis. JL and YZ contributed to sample collection, storage, and DNA sequencing. RZ, WZ, and XW contributed samples/reagents. XD drafted the manuscript. XD, JL, and YC analyzed data and wrote the manuscript, with input from VAF. All authors approved the final manuscript.

## Conflict of interest

The authors declare that they have no conflict of interest.

## For more information

i      https://www.who.int/csr/don/2005_08_03/en/
ii     http://outbreaknewstoday.com/thailand-reports-340-streptococcus-suis-cases-in-2020/
iii    https://www.who.int/bulletin/volumes/88/6/09-067959/en/

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
