## [Review Process File · EMBO Molecular Medicine]

The global emergence of a novel *Streptococcus suis* clade associated with human infections

XINGXING DONG, Yanjie Chao, Yang Zhou, Rui Zhou, Wei Zhang, Vincent Fischetti, Xiaohong Wang, Ye Feng, and Jinquan Li

DOI: [10.15252/emmm.202013810](https://doi.org/10.15252/emmm.202013810)

Corresponding authors: Jinquan Li (lijinquan2007@gmail.com) , Ye Feng (pandafengye@qq.com)

Review Timeline:

Submission Date:	8th Dec 20
Editorial Decision:	13th Jan 21
Revision Received:	13th Apr 21
Editorial Decision:	11th May 21
Revision Received:	20th May 21
Accepted:	21st May 21

Editor: Zeljko Durdevic

Transaction Report:

13th Jan 2021

Dear Dr. Li,

Thank you for the submission of your manuscript to EMBO Molecular Medicine. We have now received feedback from the three reviewers who agreed to evaluate your manuscript. As you will see from the reports below, the referees acknowledge the interest of the study but also raise serious and partially overlapping concerns that should be addressed in a major revision.

Addressing the reviewers' concerns in full will be necessary for further considering the manuscript in our journal, and acceptance of the manuscript will entail a second round of review. EMBO Molecular Medicine encourages a single round of revision only and therefore, acceptance or rejection of the manuscript will depend on the completeness of your responses included in the next, final version of the manuscript. For this reason, and to save you from any frustrations in the end, I would strongly advise against returning an incomplete revision.

We realize that the current situation is exceptional on the account of the COVID-19/SARS-CoV-2 pandemic. Therefore, please let us know if you need more than three months to revise the manuscript.

I look forward to receiving your revised manuscript.

Yours sincerely,

Zeljko Durdevic

***** Reviewer's comments *****

Referee #1 (Comments on Novelty/Model System for Author):

More studies using Zebrafish model to pinpoint the genetic background for the virulence of HAC are needed.

Referee #1 (Remarks for Author):

366 S. suis isolates from human or pigs collected in 2015-16 were sequenced. Comparative genomic analysis was done with 1634 genomes. Three clades were found, and most human isolates belonged to one clade, which refers to an HAC.

In general, I feel this is a pure comparative genomics study, that provides some new information to the molecular epidemiology of *S. suis* infection.

I have the following comments to this article:

1. P. 5, lines 116-7. 44 isolates from healthy pigs also belonged to the HAC clade. The observation can not be inferred to "a possible high virulence potential" of the isolates.
2. P. 6. Few isolates were randomly picked up for virulence test using a Zebrafish model. The evidence is very preliminary and limited. If possible, more isolates should be tested to get a more solid conclusion.
3. P. 9-10. A few genes were identified and implicated these genes contribute to the virulence of human isolates. The evidence for such description is weak, although some references were cited here. Some studies cited appeared fragmented.
4. The bacterial isolates sequenced in this study appeared outdated. A deeper and wider look at the longitudinally collected isolates from different sources may provide a more comprehensive view on the molecular epidemiology of *S. suis*.

Referee #2 (Remarks for Author):

The manuscript is well written and attractive for many basic researchers.
The referee requests followings to be convinced.

1. The word "evolution" should be defined thorough this manuscript. Sometimes diversification or adaptation appears more appropriate instead of evolution. The authors assume the ability to infect human is evolution for the bacterium but this is not equal to evolution.

2. The bacterium genome is recombinogenic as below.

<https://www.nature.com/articles/ncomms7740>

Therefore, the analysis such as FastGEAR which can detect mosaicism in bacterial genomes should improve the quality of this manuscript.

3. The authors discuss clade-specific genes and genomic regions. Therefore, bacterial GWAS based on SNPs and genes will deepen the findings of the manuscript.

Please see below for the analyses.

<https://pubmed.ncbi.nlm.nih.gov/33033372/>

<https://pubmed.ncbi.nlm.nih.gov/32518186/>

Other points. Please check the value through the manuscript.

L93-94 "the BAPS7 group contains 96% (549/570) human isolates (Supplementary Figure 1A-B)"

Please see figS1B again. It does not appear 96%.

L94-95 "including all the highly virulent isolates of sequence type 1 (ST1) and the epidemic isolates of ST7. Please see figS1C again. It does not appear ST7.

Referee #3 (Comments on Novelty/Model System for Author):

This manuscript presented the evolution of human-associated *S. suis* by using the population genomics approaches. No model system was used in this manuscript.

Referee #3 (Remarks for Author):

This manuscript by Dong et al. presented a nice evolution analysis of one hundred *Streptococcus suis* isolates collected from human patients by using the population genomics approach. *S. suis* has been thought to be restricted in animals, say, pigs; however, several outbreaks of *S. suis* recently emerged in China. The natural transmit mechanism of this important zoonotic pathogen between pigs and humans is unclear; the genomic evolution described in this manuscript does provide new and crucial insight. By using WGS-based phylogenetic analysis, this study has identified an evolutionary cluster of human-associated *S. suis* strains, and then proposed the evolutionary path/origin. Besides, this manuscript proposed a diagnostic method to identify pathogenic *S. suis* strains, providing a useful surveillance tool. This manuscript is written clearly. I have only a few suggestions listed as follows.

Major:

1. Recombination, which is frequent in many bacterial species, is known to be able to distort the topology of the phylogenetic tree. The phylogeny with long branches (Fig. 1) implies recombination is abundant in the dataset. Did the authors consider the effect of this occurring on the population level?
2. Please include a PCR experiment on Clade A-specific markers and give reference to the sequence of the target genes or markers identified.
3. In general, population genomic analysis with publicly available sequences sometimes has the problem of sampling bias. Have the authors considered or addressed this by downsampling or use of alternative approaches that may be less prone to sampling bias?
4. WGS quality control would be described. Why BM407 was employed as the reference genome in this study?
5. In Figure 3, how to define the major lineages? And how to determine the PAI present or absent in one isolate?

Minor:

1. Figure 1b, the scale would be provided.
2. Page 4, Line 88, "Genbank" would be "GenBank".
3. Page 16, Line 358, "Sequence homology" would be "Sequence identities".
4. Page 18, Line 399, the reference of ICEberg would be the paper of Liu, et al. (Nucleic Acids Research, 47(D1):D660-D665. doi: 10.1093/nar/gky1123).
5. Page 18, Line 405, "BLSTN" would be "BLASTN".
6. Page 19, Line 419, the authors would check the license of SPSS. If not, please use the R package to do the Kaplan-Meier analysis.

**** Reviewer's comments ****

Referee #1 (Comments on Novelty/Model System for Author):

More studies using Zebrafish model to pinpoint the genetic background for the virulence of HAC are needed.

Reply: As suggested by this reviewer, we have performed additional infection experiments with the Zebrafish model. Please see the detailed results and our reply below.

Referee #1 (Remarks for Author):

366 *S. suis* isolates from human or pigs collected in 2015-16 were sequenced. Comparative genomic analysis was done with 1634 genomes. Three clades were found, and most human isolates belonged to one clade, which refers to an HAC. In general, I feel this is a pure comparative genomics study, that provides some new information to the molecular epidemiology of *S. suis* infection.

I have the following comments to this article:

1. P. 5, lines 116-7. 44 isolates from healthy pigs also belonged to the HAC clade. The observation can not be inferred to "a possible high virulence potential" of the isolates.

Reply: We thank this reviewer for pointing out this logic error in our writing. We have revised the relevant sentences and explained our rationale in another way. The modified texts are as below:

To understand whether these human-associated strains possess higher virulence potential, we have randomly selected 14 isolates in HAC and tested their pathogenesis in an established infection model for *S. suis* (Zaccaria et al, 2016). In addition, we have tested ten healthy-pig isolates from HPC, and strain SC19 responsible for the 2005 human outbreak as a positive control. As depicted in Figure 2B, all 14 isolates from HAC displayed a significantly higher mortality than the HPC isolates ($P < 0.0001$, Fisher's exact test). As expected, the positive control SC19 caused a high degree of mortality in this model of infection (Fig 2B, Appendix Table S1), very similar to the new HAC strains tested. Notably, five of these HAC strains were originally isolated from healthy pigs, indicating that healthy pigs may be a reservoir of HAC strains with high virulence potential in human.

2. P. 6. Few isolates were randomly picked up for virulence test using a Zebrafish model. The evidence is very preliminary and limited. If possible, more isolates should be tested to get a more solid conclusion.

Reply: Following the reviewer's suggestion, we have doubled the isolates tested in our validation experiment, by including another ten strains for HAC and HPC clades. The new data are fully consistent with our previous results, further corroborating our hypothesis. Thus far, we have tested 25 isolates in total (Appendix Table S1), including 15 HAC isolates (SC19 as control) and 10 HPC isolates. The new data are summarized in the updated Figure 2 in the main text, which makes the conclusion much stronger thanks to this reviewer.

3. P. 9-10. A few genes were identified and implicated these genes contribute to the virulence of human isolates. The evidence for such description is weak, although some references were cited here. Some studies cited appeared fragmented.

Reply: As a large-scale omics study, our genomic comparisons between the isolates of different virulence indicated a number of putative virulence genes. Due to the limited understanding of *S. suis* pathogenesis in human, we try to be conservative and not to over interpret these predictions, especially in the absence of well-controlled genetic experiments in wet-labs. However, a few of these genes have already been demonstrated to

Table 1 The recombination analysis based on 7 *S. suis* housekeeping genes among three clades we identified.

Clade	R/theta	1/delta	nu	r/m
Human-associated clade (HAC)	0.793814	0.004326	0.043096	7.91
Diseased-pig clade (DPC)	0.14728	0.002268	0.017389	1.13
Healthy-pig clade (HPC)	19.5343	0.010018	0.030778	60.02

The formula $r/m=R/\theta*\delta*\nu$ was used for compute r/m according to ClonalFrameML.

3. The authors discuss clade-specific genes and genomic regions. Therefore, bacterial GWAS based on SNPs and genes will deepen the findings of the manuscript. Please see below for the analyses.

<https://pubmed.ncbi.nlm.nih.gov/33033372/>

<https://pubmed.ncbi.nlm.nih.gov/32518186/>

Reply: Following this suggestion, we have performed GWAS analysis with GEMMA software to search for genes and SNPs that are tightly associated with the host phenotype. At the level of gene presence/absence, 66 genes met the criteria, 25 of which also appeared in our previous analysis of host-related genes (Appendix Figure S7, Appendix Table S5). The strong agreement between these two methods shows the reliability of our analyses. At the level of SNPs, we have identified 71 SNPs that are related with host range. 17 of them are missense mutations. As indicated below, the genes carrying these SNPs encode functions enriched in “metabolism” and “cellular processes and signaling” (Appendix Figure S10, Appendix Table S7). These data are included in the supplemental files in the revised manuscript.

Other points.

Please check the value through the manuscript. L93-94 "the BAPS7 group contains 96% (549/570) human isolates (Supplementary Figure 1A-B)" Please see figS1B again. It does not appear 96%.

Reply: We apologize for the ambiguity of our writing. We meant that 96% of the human isolates fell into the BAPS7 group. The relevant sentence has been revised for clarity.

L94-95 "including all the highly virulent isolates of sequence type 1 (ST1) and the epidemic isolates of ST7. Please see figS1C again. It does not appear ST7.

Reply: FigS1C shows serotyping results instead of MLST results.

Referee #3 (Comments on Novelty/Model System for Author):

This manuscript presented the evolution of human-associated *S. suis* by using the population genomics approaches. No model system was used in this manuscript.

Referee #3 (Remarks for Author):

This manuscript by Dong et al. presented a nice evolution analysis of one hundred *Streptococcus suis* isolates collected from human patients by using the population genomics approach. *S. suis* has been thought to be restricted in animals, say, pigs; however, several outbreaks of *S. suis* recently emerged in China. The natural transmit mechanism of this important zoonotic pathogen between pigs and humans is unclear; the genomic evolution described in this manuscript does provide new and crucial insight. By using WGS-based phylogenetic analysis, this study has identified an evolutionary cluster of human-associated *S. suis* strains, and then proposed the evolutionary path/origin. Besides, this manuscript proposed a diagnostic method to identify pathogenic *S. suis* strains, providing a useful surveillance tool. This manuscript is written clearly. I have only a few suggestions listed as follows.

Major:

1. Recombination, which is frequent in many bacterial species, is known to be able to distort the topology of the phylogenetic tree. The phylogeny with long branches (Fig. 1) implies recombination is abundant in the dataset. Did the authors consider the effect of this occurring on the population level?

Reply: As pointed out by the reviewer 2 as well, recombination is frequent in bacteria such as streptococci. We have considered the possibility that recombination might distort the topology of the phylogenetic tree, and finally excluded the impact of recombination on our results. Using either chromosomal alignment or recombination-free alignment (using Gubbins to filter recombination) as input, two trees constructed showed the same topology, with only minor difference in branch length (Fig 1A, Appendix Figure S2).

2. Please include a PCR experiment on HAC-specific markers and give reference to the sequence of the target genes or markers identified.

Reply: We have performed PCR experiment on two HAC-specific marker genes (Appendix Figure S12-13). Their sequences are now provided in Appendix Table S10.

3. In general, population genomic analysis with publicly available sequences sometimes has the problem of sampling bias. Have the authors considered or addressed this by downsampling or use of alternative approaches that may be less prone to sampling bias?

Reply: Yes, we have tried down-sampling to eliminate potential biases. Out of 562 strains, we have manually selected 174 representative isolates based on sampling time and geographic location, and repeated phylogeographic analysis. As the result, the phylogenetic tree after down-sampling also identified three major Asian lineages, showing very similar topological structure to the original tree based on all isolates (Appendix Figure S4). Thus we believe that that sampling bias has little impact on our final conclusions.

In addition, the Bayesian analysis of population structure revealed that there was no significantly biased population genetic structure in the distribution of sampling locations except Vietnam. Moreover, isolates from China or United Kingdom were distributed in all nine BAPS groups mentioned in our manuscript (Appendix Figure S1). In particular, the human-associated clade that we identified contains isolates from 10 countries, which further verifies the illustrative nature of our analysis.

4. WGS quality control would be described. Why BM407 was employed as the reference genome in this study?

Reply: We have now added the method for the genome quality control, as well as the QC statistics for the genome assemblies (in Appendix Table S13).

BM407 is a recommended reference genome in the NCBI database. This isolate belongs to ST1 by MLST, one of the most prevalent and virulent clones worldwide. In addition, because our analysis is based on core genome, the choice of reference genomes will not significantly affect the analyses and conclusions.

5. In Figure 3, how to define the major lineages? And how to determine the PAI present or absent in one isolate?

Reply: Three major lineages were determined based on the topological structure identified by phylogeny analysis. We more focused on three Asian lineages because of their high occurrence in human infections and three past outbreaks of in human in Asia.

To determine PAI in clinical isolates, MGEs candidates were firstly located by comparative genomic analysis using MAUVE v2.4.0. Secondly, PAI was predicted among MGE candidates using VRprofile v2.0.

Minor:

1. Figure 1b, the scale would be provided.

Reply: The scale has been added to Figure 1B.

2. Page 4, Line 88, "Genbank" would be "GenBank".

3. Page 16, Line 358, "Sequence homology" would be "Sequence identities".

4. Page 18, Line 399, the reference of ICEberg would be the paper of Liu, et al. (Nucleic Acids Research, 47(D1):D660-D665. doi: 10.1093/nar/gky1123).

5. Page 18, Line 405, "BLSTN" would be "BLASTN".

Reply: All of the above corrected.

6. Page 19, Line 419, the authors would check the license of SPSS. If not, please use the R package to do the Kaplan-Meier analysis.

Reply: We have used SPSS with license code:

VDOV7M8KUEIAWBZIKPP6DUKX4JIO3LWRSJQW4BTDCU5NS28ZLZSSROOZQ8HASZ6VUH
RZRZ8I8DGWIFY9WJTIRD5P9Y

11th May 2021

Dear Dr. Li,

Thank you for the submission of your revised manuscript to EMBO Molecular Medicine. I am pleased to inform you that we will be able to accept your manuscript pending the following final amendments:

1) In the main manuscript file, please do the following:

- Correct/answer the track changes suggested by our data editors by working from the attached/uploaded document.
- Remove text highlight colour.
- Remove "data not shown" (p. 9).
- Add figure callouts for Fig1A and B and for Fig3A and B.
- Make sure that all special characters display well.
- Rename the "Financial support" section to "Acknowledgements".
- Use initials for author contributions.
- Rename "Potential conflicts of interest" to "Conflict of interest".
- Indicate in legends exact $n=$ and exact $p=$ values, not a range, along with the statistical test used. To keep the figures "clear" some authors found providing an Appendix table Sx with all exact p -values preferable. You are welcome to do this if you want to.
- Correct the reference citation in the reference list. Where there are more than 10 authors on a paper, 10 will be listed, followed by "et al.". Please check "Author Guidelines" for more information. <https://www.embopress.org/page/journal/17574684/authorguide#referencesformat>
- In addition to the accession number please provide URL for deposited datasets. Use the following format to report the accession number of your data:

[data type]: [full name of the resource] [accession number/identifier] ([doi or URL or identifiers.org/DATABASE:ACCESSION])

Please check "Author Guidelines" for more information.

<https://www.embopress.org/page/journal/17574684/authorguide#availabilityofpublishedmaterial>

- 2) Appendix: Please add tables 1, 3, 4, 5, 6, 7, 8, 10 to Appendix, relabel them to "Appendix Table S1" etc., and list them in table of content. Also, correct the callouts of the tables in the text.
- 3) Dataset: Please upload tables 2, 9, 11, 12, 13 as separate dataset excel files with their legends added directly to the respective files in a separate tab. Rename the tables to "Dataset EV1" etc and accordingly correct their callouts in the text.
- 4) Synopsis image: Some of the used fonts are too small, please simplify the image and increase the font size so that the text is readable when the image is resized to 550-px width and 400-px height.
- 5) For more information: There is space at the end of each article to list relevant web links for further consultation by our readers. Could you identify some relevant ones and provide such information as well? Some examples are patient associations, relevant databases, OMIM/proteins/genes links, author's websites, etc...
- 6) As part of the EMBO Publications transparent editorial process initiative (see our Editorial at <http://embomolmed.embopress.org/content/2/9/329>), EMBO Molecular Medicine will publish online a Review Process File (RPF) to accompany accepted manuscripts. This file will be published in

conjunction with your paper and will include the anonymous referee reports, your point-by-point response and all pertinent correspondence relating to the manuscript. Let us know whether you agree with the publication of the RPF and as here, if you want to remove or not any figures from it prior to publication. Please note that the Authors checklist will be published at the end of the RPF. 7) Please provide a point-by-point letter INCLUDING my comments as well as the reviewer's reports and your detailed responses (as Word file).

I look forward to reading a new revised version of your manuscript as soon as possible.

Yours sincerely,

Zeljko Durdevic

***** Reviewer's comments *****

Referee #2 (Comments on Novelty/Model System for Author):

I believe the authors added enough 'wet experiments' based on the other referees.

Referee #2 (Remarks for Author):

The reviewer sincerely appreciates the authors effort to answer previous comments. Now the manuscript is satisfactory as a high quality manuscript in bacterial genomics.

Referee #3 (Remarks for Author):

My comments had been considered and the quality of this revised manuscript was enhanced. In my opinion, this revision might satisfy the standards of EMBO Molecular Medicine.

****** Reviewer's comments ******

Referee #2 (Comments on Novelty/Model System for Author):

I believe the authors added enough 'wet experiments' based on the other referees.

Referee #2 (Remarks for Author):

The reviewer sincerely appreciates the authors effort to answer previous comments. Now the manuscript is satisfactory as a high quality manuscript in bacterial genomics.

Referee #3 (Remarks for Author):

My comments had been considered and the quality of this revised manuscript was enhanced. In my opinion, this revision might satisfy the standards of EMBO Molecular Medicine.

Reply: We thank both reviewers for their appraisals.

We are pleased to inform you that your manuscript is accepted for publication and is now being sent to our publisher to be included in the next available issue of EMBO Molecular Medicine.

Corresponding Author Name: Jinquan Li; Ye Feng

Manuscript Number: EMM-2020-13810